# Effects of Different Feeding Methods on Growth Performance, Enzyme Activity, Rumen Microbial Diversity and Metabolomic Profiles in Yak Calves

**DOI:** 10.3390/microorganisms14010081

**Published:** 2025-12-30

**Authors:** Hongli Wang, Wanhao Ma, Muhammad Irfan Malik, Ali Mujtaba Shah, Aixin Liu, Guangwei Hu, Jianwu Jing, Hongkang Li, Yayu Huang, Qunying Zhang, Jianwei Zhou, Binqiang Bai, Yingkui Yang, Zhenqun Wang, Jianbo Zhang, Lizhuang Hao

**Affiliations:** 1Key Laboratory of Plateau Grazing Animal Nutrition and Feed Science of Qinghai Province, Qinghai University, Xining 810016, China; whl236789qhu@foxmail.com (H.W.); mawanh93@126.com (W.M.); aixinliu259@foxmail.com (A.L.); zqy1010307564@126.com (Q.Z.); binqiangbai@163.com (B.B.); yykui@qhu.edu.cn (Y.Y.); mkywzhq@126.com (Z.W.); 2Qinghai Yak Breeding Extension Service Center, Datong 810100, China; qhhgw@sina.com (G.H.); qmzyjjw@163.com (J.J.); 66181323@qq.com (H.L.); 3Department of Veterinary Sciences, University of Turin, 10095 Turin, Italy; dr.irfan279@gmail.com; 4Guangdong Provincial Key Laboratory of Animal Nutrition and Regulation, College of Animal Science, South China Agricultural University, Guangzhou 510642, China; alimujtabashah@scau.edu.cn; 5PEGASE, INRAE, Institut Agro, 35590 Saint-Gilles, France; yayu.huang.fr@gmail.com; 6State Key Laboratory of Grassland Agro-Ecosystems, Center for Grassland Microbiome, College of Pastoral Agriculture Science and Technology, Lanzhou University, Lanzhou 730000, China; zhoujw@lzu.edu.cn

**Keywords:** yak calves, growth performance, rumen fermentation, rumen microbiome structure, rumen microbial metabolome

## Abstract

Yaks are important for the ecology and economy of the Qinghai-Tibetan Plateau. The growth of the yak industry depends on sustainable and accelerated growth of calves, sustaining herd reproduction and production systems. Yak calves born in the summer months of June and July are faced with a heightened risk of winter mortality. Exclusive traditional, natural feeding prolongs the suckling period, and this leads to a series of problems due to the harsh high-altitude environment, such as inadequate nutrition leading to retarded growth and an imbalanced herd structure. To enhance growth performance and breeding efficiency, 12 male calves of similar weights (68.53 ± 6.41 kg) were randomly assigned to a control group (suckle the dam (SU)) or an experimental group (early weaning with full feeding, concentrate and oat hay at a 7:3 ratio (CO)). The results showed that compared with suckling, early weaning with full feeding significantly improved the growth performance, volatile fatty acids and digestive enzyme activity. The abundance of the Firmicutes was reduced, but there was an increased abundance of Bacteroidetes, which affected the rumen metabolome. In conclusion, early weaning with full feeding improves growth performance, promotes rumen fermentation and carbohydrate degradation, reduces the diversity and richness of rumen microbial flora and alters the content and pathways of metabolites in yak calves. These factors contribute to the growth and market readiness of yak calves born in June and July, accelerate herd turnover and enhance the production efficiency of grazing yaks.

## 1. Introduction

Yak (*Bos grunniens*) is an endemic species of the Qinghai-Tibetan Plateau, growing up in areas 3000~6000 m in altitude. As a unique type of livestock among large herbivores, it is well adapted to the harsh plateau climate and capable of efficiently utilizing forage resources from alpine grasslands. They have strong cold resistance, hunger tolerance and disease resistance, and they can survive and grow well in harsh environment conditions [1]. Yaks play a crucial role in the plateau’s livestock systems, supplying meat, milk, hides, fiber and dung used as fuel. These outputs constitute a major source of both economic sustenance and daily necessities for local communities, providing essential resources that other animals cannot provide in such harsh conditions [2]. The future of the yak industry largely depends on a well-managed herd renewal, as the quality of adult cows and bulls strongly relies on the successful weaning and early growth of calves [3]. Early life feeding is a critical period and could have long term effects on udder development and, consequently, milk production in proceeding lactation. Accelerating growth by providing high-energy concentrate not only reduces the age of sexual maturity but also lowers the time period of attaining the age of first calving [4,5].

Due to poorly established commercial agricultural systems in the Qinghai-Tibetan Plateau, most yaks are still reared on an ancient feeding system. The traditional weaning age of yak calves is quite late, typically occurring around 1.5–2 years of age under extensive grazing conditions [6]. Yak calves suckle their mothers while grazing in natural pastures, and this prolonged suckling period aligns with the harsh environmental conditions and nutritional challenges of the high-altitude Qinghai-Tibetan Plateau, where yaks are commonly raised [7]. Prolonged suckling delays rumen development, whereas increased solid feed intake accelerates it. Early weaning calves with a higher solid feed intake show more advanced rumen epithelial, papillary and muscular growth [8,9,10,11]. Adopting appropriate feeding and management strategies can therefore shorten the production cycle of yaks and enhance calf growth performance. Early feeding strategies and diet compositions strongly influence rumen development in young ruminants [12]. Adjusting the roughage-to-concentrate ratio and feeding management modify rumen microbial diversity and abundance, thereby shaping host–microbiome interactions and promoting rumen development [13]. High-concentrate and roughage intake stimulate rumen tissue growth, microbial proliferation and volatile fatty acid (VFA) production, improving overall productivity [14,15].

As pastures gradually enter the period of withering and yellowing, yak calves born between June and July are affected by the foraging quality and the nutritional status of their mothers, exhibiting a poorer physical condition and overwintering ability compared with those born between March and April and resulting in higher rates of weak calves and mortality. Direct sales yield low prices, early weaning with a milk replacer and starter feed incurs high costs, and extended nursing reduces the turnover rate of mother yaks [16]. The rumen, as the primary digestive organ in ruminants, harbors a vast community of microorganisms that play a crucial role in an animal’s growth. Rumen microbes form a complex symbiotic network with the host animal, playing a vital role in rumen fermentation, feed degradation and overall host health [17,18]. However, the functional and metabolic characteristics of rumen microorganisms in weaned calves, particularly those grazing in pastoral regions, remain poorly understood. Current management practices may have implications on rumen fermentation and microbial colonization. This study aims to collect information regarding the rumen microbial community structure, VFA concentrations and metabolite profiles of suckled and high concentrate-fed yak calves, evaluating the effects of different feeding methods on yak calves after six months of age. This study aims to explore a yak calf rearing model that enables timely weaning and rapid finishing, thereby accelerating herd turnover and enhancing the production efficiency of grazing yaks.

## 2. Materials and Methods

### 2.1. Experimental Design

The experiment was conducted between November 2023 and April 2024 at the Yak Breeding Center in Datong County, Qinghai Province, located at an average altitude above 3000 m, with an average temperature and humidity of −2.50 °C and 46.96% during the experiment, respectively. Twelve 6-month-old male yak calves with similar body weights (68.53 ± 6.41 kg) that were in good health were selected for the study. All experimental animals were vaccinated and dewormed in accordance with the immunization protocols of the Qinghai Yak Breeding and Extension Service Center. Calves were evenly assigned to two treatments (6 yaks/treatment). The control calves were allowed to suckle the dam (SU). They could suckle ad libitum and had access to the same forage consumed by their dams, while the calves in the experimental group were fed a green oat hay-based total mixed ration with concentrate (CO) (forage, 70:30) under group-housed feeding conditions. In addition, the suckling calves and their mothers were also under group-housed feeding, with no additional supplementation provided to the calves in the SU group. Both groups were allowed ad libitum access to water. The trial included a 15-day pre-experimental adaptation period followed by 120 days for data collection. The calves in the CO group were fed the experimental diet twice a day at 8:30 a.m. and 5:30 p.m., and the daily feed ration for the calves was formulated at a rate of 2.5–3% of their live body weights.

### 2.2. Experimental Diet

The composition and chemical analysis of the diet fed to the CO yak calves are provided in Table 1. The nutritional composition of the concentrate-fed female yaks was as follows: metabolizable energy 9.88%, crude protein 12.26%, ether extract 2.89%, ash 4.36%, neutral detergent fiber 15.80%, acid detergent fiber 8.50%, phosphorus (P) 0.65% and calcium (Ca) 0.09%. Each female yak was provided 6 kg of oat hay and 1.5 kg of concentrate. During feeding, the chopped hay was provided first, with the concentrate portion of the diet placed on top. Additionally, the measured milk quality indicators of the SU group’s female yaks were as follows: milk fat 2.50%, non-fat milk solids 10.68%, lactose 5.87% and protein 3.92% [16].

### 2.3. Determination of Growth Performance

On the first and 120th day of the experiment, all calves were weighed in the morning after an overnight fast and before feeding with an electric weighing balance. Additionally, their body heights, body lengths and chest circumferences were measured. The average daily gain (ADG) was calculated as (final body weight − initial body weight)/120 days.

### 2.4. Sample Collection

After completion of the experiment, the yak calves were slaughtered. During the slaughtering process, we strictly adhered to animal ethical standards (approval number: PJ202401-68). Electrical stunning was used to euthanize the animals humanely. Then, rumen fluid was collected within 10 min, and the samples were filtered through four layers of gauze and aliquoted into 5 mL cryotubes. These aliquots were immediately snap-frozen in liquid nitrogen and subsequently transferred to a laboratory, where they were stored at −80 °C for subsequent analysis. One portion was preserved for measuring VFA and enzyme activity, while the other portion was used for 16S rRNA sequencing and metabolite profiling of gastric microorganisms.

### 2.5. Determination of Rumen Fluid VFAs

The rumen fluid sample stored at −80 °C was thawed at room temperature. A 5 mL aliquot was transferred into a centrifuge tube and centrifuged at 15,000 rpm at 4 °C for 15 min. Subsequently, 1 mL of the supernatant was pipetted into a 1.5 mL centrifuge tube and mixed with 0.2 mL of 25% metaphosphoric acid solution. Furthermore, the internal standard was prepared using 0.2 mL of 25% metaphosphoric acid solution and supplemented with 1.0 mL, 0.8 mL, 0.6 mL, 0.4 mL and 0.2 mL of VFA mixed standard solution (Shanghai Macklin Biochemical Co., Ltd., Shanghai, China) along with 0 mL, 0.2 mL, 0.4 mL, 0.6 mL and 0.8 mL of ultrapure water, respectively. Mixed standard solutions of different concentrations were contained and incubated in an ice water bath for 1 h. After incubation, the mixture was centrifuged again at 15,000 rpm for 30 min at 4 °C. The resulting supernatant was collected using a 2 mL syringe, filtered through a 0.22 μm membrane and transferred into brown vials (V > 0.5 mL) for analysis. The concentration of VFAs in the rumen fluid samples was analyzed in triplicate for each cow. The concentrations of VFAs in the rumen fluid were quantified using a gas chromatograph (Shimadzu, GC-2014C, Kyoto, Japan) equipped with an AT-FFAP capillary column (30.0 m × 0.32 mm × 0.32 μm), and the peak time for the VFAs was between 7 and 12 min. A micro-syringe was used to inject 0.6 μL of the sample into the injection port (VFA mixed solution standard curve R^2^ > 0.99). The chromatographic conditions were as follows. Nitrogen was used as the carrier gas in constant current mode at a flow rate of 2.1 mL/min, with a split ratio of 40:1, an average linear velocity of 38 cm/s and a column pressure of 11.3 psi (0.1 MPa). The injector and detector temperatures were both set at 250 °C.

### 2.6. Enzyme Activity Determination

The activities of digestive enzymes and metabolic enzymes with VFA production were measured using a one-step sandwich ELISA with dual antibodies (Psenno Biotechnology Co., Ltd., Shanghai, China). Samples, standards and HRP-labeled detection antibodies were sequentially added to the microplate wells pre-coated with the specific enzyme antibody, followed by incubation and thorough washing. The substrate 3,3′,5,5′-Tetramethylbenzidine (TMB) was used for color development. Under peroxidase catalysis, TMB turns blue and then changes to yellow after the addition of an acid. The color intensity is positively correlated with the amount of enzyme present in the sample. The absorbance (OD value) was measured at 450 nm using an ELISA microplate reader (Sunrise, F50, Männedorf, Switzerland), and the enzyme activity in the samples was calculated accordingly.

### 2.7. 16S rRNA Gene Sequencing

The DNA was extracted from the rumen fluid of 12 yaks (6/treatment) using TRIzol reagent (Invitrogen, Carlsbad, CA, USA), following the manufacturer’s instructions. The DNA concentration and purity were measured using a Qubit fluorometer, and integrity was assessed using 2% agarose gel electrophoresis. Qualified DNA samples were used as templates to amplify the V3–V4 hypervariable regions of the 16S rRNA gene with the primers 341F (5′-CCTAYGGGRBGCASCAG-3′) and 806R (5′-GGACTACNNGGGTATCTAAT-3′). Each 15 µL PCR reaction contained Phusion High-Fidelity PCR Master Mix, 0.2 µM of each primer and 10 ng of the template DNA. The PCR conditions were as follows: initial denaturation at 98 °C for 1 min, 30 cycles of denaturation at 98 °C for 10 s, annealing at 50 °C for 30 s and extension at 72 °C for 30 s, followed by a final extension at 72 °C for 5 min. The PCR products were purified using magnetic beads, pooled in equimolar ratios and examined for target band recovery. Purified amplicons were used for library construction, quantified with Qubit and real-time PCR and analyzed for their fragment size distribution using a bioanalyzer before being sequenced on an Illumina NovaSeq 6000 platform (San Diego, CA, USA). During the experiment, contamination control was implemented as follows. A negative control (blank control) and a positive control (species-standard sample) were included. Paired-end reads were assigned to corresponding samples based on their unique barcodes, after which the barcode and primer sequences were trimmed. The reliability of the amplicon sequencing data was considered acceptable when the number of sequences from the negative control was less than 1% and the proportion of reads from the positive control exceeded 90%. Paired-end reads were merged using the FLASH (version 1.2.11) analysis tool, and the final number of reads of the samples were ranged from 84,546 to 110,759.

### 2.8. Metabolite Determination

Non-targeted metabolomics analysis of the rumen fluid was carried out based on liquid-mass combination (LC-MS) technology [20,21], including sample metabolite extraction and LC-MS/MS detection, by Beijing Novogene Technology Company Limited (Beijing, China). The sample injection volume was 5 μL in positive ion mode and 8 μL in negative ion mode. The chromatographic separation was performed on a Hypersil Gold C18 column maintained at 40 °C, with a flow rate of 0.2 mL/min. The mobile phase consisted of 0.1% formic acid in water (A) and methanol (B). Mass spectrometric analysis was conducted under the following conditions. Full scans were acquired over the *m*/*z* range of 100–1500 using an electrospray ionization (ESI) source with a spray voltage of 3.5 kV. The sheath gas and auxiliary gas flows were set at 35 psi and 10 L/min, respectively. The ion transfer tube temperature was maintained at 320 °C, and the auxiliary gas heater temperature was set to 350 °C. The RF lens value was set at 60 V. Data-dependent acquisition (DDA) was employed for MS/MS scanning. The normalization method of the original data according to the following formula to obtain the relative peak areas: sample raw quantitation value/(sum of sample metabolite quantitation value/sum of QC1 sample metabolite quantitation value). Metabolite identification was supported by database searching against KEGG (https://www.genome.jp/kegg/pathway.html, 30 June 2024), and 1010 metabolites were identified.

### 2.9. Data Statistical Analysis

The original data were subjected to statistical analysis with SPSS 21.0. The growth performance and VFA data were analyzed with an independent samples *t*-test, where a *p* value < 0.05 was considered to show a significant difference. Paired-end reads were assigned to their respective samples by trimming the barcode and primer sequences, followed by merging using FLASH (version1.2.11). High-quality clean tags were obtained using Fastp (version 0.23.1), which were then aligned against the Silva database (16S; https://www.arb-silva.de/, 10 July 2024) to generate effective tags. Denoising of the effective tags was performed with DADA2 in QIIME2 (version 202202) to yield initial amplicon sequence variants (ASVs), and species annotation was conducted using the Micro_NT database. Multiple sequence alignment was carried out in QIIME2 to examine the phylogenetic relationships of each ASV. The sample with the fewest sequences was used as the reference for rarefaction. The top 10 taxa at the phylum and genus levels were selected to generate relative abundance distribution histograms using Perl’s SVG function. A heatmap illustrating the top 35 taxa across taxonomic ranks was created in R (version 4.0.3). Diversity analyses, including alpha and beta diversity, were computed using QIIME2 and R. Association analysis and functional prediction were performed using relevant R packages.

For metabolomic data analysis, the raw data files obtained from mass spectrometry were imported into Compound Discoverer 3.3 (hereafter referred to as CD 3.3) for spectral processing and database searching to obtain the qualitative and quantitative results of the metabolites. Quality control was then performed to ensure the accuracy and reliability of the data. Metabolites with a coefficient of variation (CV) below 30% in the quality control (QC) samples were retained as the final identification results. Differential metabolites were screened using the criteria of VIP > 1.0, fold change (FC) > 1.5 or FC < 0.667 and a *p* value < 0.05. Based on the resulting dataset, systematic pathway annotation was performed for all identified metabolites, and differential analysis along with KEGG pathway analysis was conducted for all differential metabolites. Data transformation and processing in metabolomics were carried out using Meta X software (http://metax.genomics.cn, 30 June 2024). Principal Coordinates Analysis (PCoA), partial least squares-discriminant analysis (PLS-DA), cluster analysis and volcano plot visualization were performed using Python (version 3.5.0) and R (version 4.0.3). KEGG enrichment analysis visualized by bubble charts was conducted using Python (3.5.0) and R (4.0.3).

## 3. Results

### 3.1. Comparison of Feeding Methods on the Growth Performance of Yak Calves

The initial wither heights, body lengths, chest circumferences and weights of the yak calves were similar (*p* > 0.05) in both the SU and CO groups (Table 2). At the end of the experiment, the final wither heights and chest circumferences were higher (*p* < 0.05) in the CO group compared with the SU calves, while the body lengths were similar (*p* > 0.05) in both treatments. The final body weights were higher (*p* < 0.05) for the CO yaks compared with the SU calves, and the final weights of the CO yaks were 33.28% greater than those of the SU yaks. Similarly, the total weight gain and ADG were higher (*p* < 0.05) in the CO group compared with the SU group.

### 3.2. Effects of Different Feeding Methods on Rumen VFA Content of Yak Calves

The rumen fluid in the CO group exhibited significantly higher (*p* < 0.05) concentrations of acetate, propionate, butyrate and total volatile fatty acids compared with the SU group (Figure 1A–D), while those of isobutyrate, isovalerate, valerate, acetate and propionate were not influenced (*p* > 0.05) by the feeding method (Appendix A).

### 3.3. Effects of Different Feeding Methods on Enzyme Activities of Yak Calves

Among the digestive enzymes assayed, the activities of cellulase, hemicellulase, amylase, pepsin, sucrase and trypsin were higher in the CO yaks compared with the SU yak calves (*p* < 0.05) (Figure 2A–C, Appendix A). Among the metabolic enzymes analyzed, the activities of succinate dehydrogenase, butyrate kinase, phosphofructokinase,, pyruvate kinase were significantly higher in the CO group compared to the SU group (*p* < 0.05) (Figure 2D, Appendix A). No significant differences were observed between the two groups in the activities of ferredoxin oxidoreductase, and Na^+^/K^+^-ATPase (*p* > 0.05) (Appendix A).

### 3.4. Analysis of Rumen Microbial Community Composition

Statistical analysis of the alpha diversity index revealed that the Goods_coverage index values for both groups exceeded 0.99, indicating sufficient sample coverage (Appendix A). Based on the obtained feature sequences, common and unique ASVs across different treatment groups were analyzed. A total of 8880 ASVs were identified between the two groups, with 2124 ASVs shared commonly. The number of unique ASVs in the CO and SU groups was 2944 and 3812, respectively. Compared with the SU group, the CO group showed 14.62% fewer total ASVs and 22.77% fewer unique ASVs (Figure 3A). The alpha diversity index, represented by the Shannon and Chao 1 indices, reflects the microbial diversity and richness in the samples. Significant differences were observed between the two groups, with both the Shannon and Chao 1 indices being significantly higher (*p* < 0.05) in the SU group compared with the CO group (Figure 3B,C, Appendix A). PCoA based on unifrac distances showed clear separation between the two groups along the first principal component (PC1), which accounted for 60.9% or 21.89% of the variation, indicating significant differences in rumen microbial composition (Figure 3D,E).

At the phylum level, the dominant phyla in the rumen microbiota of both the CO and SU groups were Firmicutes (relative abundances: 45.12% and 56.43%, respectively), Bacteroidota (50.04% and 35.88%, respectively) and Euryarchaeota (2.26% and 5.11%, respectively). Compared with the SU group, the CO group showed a decrease in the relative abundances of Firmicutes and Euryarchaeota by 20.04% and 55.77%, respectively, while the relative abundance of Bacteroidota increased by 39.46% (Figure 4A, Appendix A). At the genus level, the dominant rumen microbiota genera in both groups (name of groups as CO vs. SU) were *Rikenellaceae_RC9_gut_group* (relative abundances: 10.97% and 9.27%, respectively) and *Prevotella* (8.58% and 5.56%, respectively). The relative abundance of *Prevotella* in the CO group was 54.32% higher than that in the SU group. However, with the exception of *Ruminococcus*, notable differences were observed in the abundances of other microbiota genera between the two groups. In the SU group, the subsequent dominant genera included *Methanobrevibacter* (4.85%), *Christensenellaceae_R-7_group* (4.40%), *NK4A214_group* (4.40%), *Succiniclasticum* (4.27%), *Saccharofermentans* (3.30%), *Papillibacter* (2.81%), *Ruminococcus* (2.52%) and *Prevotellaceae_UCG-001* (1.76%). In the CO group, the corresponding genera were *Christensenellaceae_R-7_group* (5.11%), *NK4A214_group* (4.47%), *Saccharofermentans* (3.05%), *Succiniclasticum* (3.04%), *Prevotellaceae_UCG-001* (2.94%), *Papillibacter* (2.77%), *Methanobrevibacter* (2.20%) and *Ruminococcus* (1.10%). Compared with the SU group, the CO group exhibited decreases in the relative abundances of *Methanobrevibacter* and *Succiniclasticum* by 54.67% and 28.66%, respectively, while *Prevotellaceae_UCG-001* and *Christensenellaceae_R-7_group* increased by 66.82% and 16.23%, respectively (Figure 4B, Appendix A).

To assess whether the differences in community structure between groups were statistically significant, identify taxa with significant abundance variations and determine the enrichment of differentially abundant species in each group, we performed LDA Effect Size (LEfSe) statistical analyses. A linear discriminant analysis (LDA) distribution bar chart illustrates taxa with an LDA score > 4, indicating biomarkers that significantly differed between groups (Figure 4C). At the phylum level, p_Bacteroidota was enriched in the CO group, while p_Firmicutes was enriched in the SU group. At the class level, c_Bacteroidia and c_Clostridia were enriched in the CO and SU groups, respectively. At the order level, o_Bacteroidales was enriched in the CO group, whereas o_Oscillospirales and o_Lachnospirales were enriched in the SU group. At the family level, f_Muribaculaceae was enriched in the CO group, while f_Ruminococcaceae and f_Lachnospiraceae were enriched in the SU group. These results indicate that Bacteroidota and Firmicutes were the key microbial phyla in the CO and SU groups.

### 3.5. Analysis of Rumen Fluid Untargeted Metabolome

Based on a *p* value <0.05, VIP > 1.0, FC > 1.5 or FC < 0.667, the metabolites identified in the two groups were categorized and subjected to statistical analysis. The results revealed that the differences in rumen liquor metabolites between the two groups were primarily concentrated in lipids and lipid-like molecules, as well as organoheterocyclic compounds. Under both negative and positive ion modes, the main classes of differential metabolites included lipids and lipid-like molecules (44.39% and 38.40%, respectively), organic acids and derivatives (16.46% and 18.00%, respectively), benzenoids (9.48% and 5.60%, respectively), organoheterocyclic compounds (9.23% and 16.00%, respectively), nucleosides, nucleotides and analogues (8.23% and 8.40%, respectively), phenylpropanoids and polyketides (6.23% and 4.00%, respectively), organic oxygen compounds (5.49% and 4.00%, respectively) and organic nitrogen compounds (0.25% and 4.80%, respectively) (Figure 5A,B).

Partial least squares discriminant analysis (PLS-DA) was employed for pairwise comparisons between the sample groups. In both positive and negative ion modes, the samples from the two groups were clearly separated. The model exhibited a high level of goodness of fit, with R2Y values approaching one, Q2Y > 0.8, and R2Y > Q2Y, indicating robust model performance (Figure 6A,B).

Volcano plots were used to compare and analyze rumen fluid metabolites between the two groups. A total of 461 metabolites were detected in the positive ion mode, and 549 were detected in the negative ion mode. Under the negative ion mode, 143 metabolites showed significant differences, among which 112 were upregulated and 31 were downregulated (Figure 6C). Notably, metabolites such as estrone, glycodeoxycholic acid, 18-β-glycyrrhetinic acid, 2-hydroxycaproic acid and gamma-glutamylmethionine were significantly upregulated, whereas 2-(3,4-dimethoxyphenyl)ethanamine, 1,3-dipyridin-3-ylpropane-1,3-dione, isoproterenol, cyclamic acid and saccharin were significantly downregulated (Appendix A). In the positive ion mode, 116 metabolites were significantly different, with 70 upregulated and 46 downregulated (Figure 6D). Metabolites such as 2′-deoxyadenosine, 4-methyl-5-thiazoleethanol, hydroxyprogesterone caproate and dehydroepiandrosterone (DHEA) were significantly upregulated, while 2-hydroxybutyric acid, N1-(2-amino-2-oxoethyl)-2-(isopropylthio)acetamide, 3-(4-methoxyphenyl)-1-methyl-1H-1,2,4-triazol-5-ol and diethyl 3-amino-6-methylthieno[2,3-b]pyridine-2,5-dicarboxylate were downregulated (Appendix A). All upregulated metabolites were present at higher concentrations in the CO group, whereas all downregulated ones were more abundant in the SU group.

These differential metabolites were subjected to Kyoto Encyclopedia of Genes and Genomes (KEGG) pathway enrichment analysis. A total of 13 metabolic pathways associated with rumen microbial differential metabolites under different feeding regimens were identified. In the negative ion mode, the differential metabolites were primarily enriched in the following pathways: lipid metabolism, amino acid metabolism, chemical structure transformation maps, nucleotide metabolism, biosynthesis of other secondary metabolites, metabolism of cofactors and vitamins and xenobiotics biodegradation and metabolism (Figure 7A). In the positive ion mode, enrichment was observed in lipid metabolism, nucleotide metabolism, amino acid metabolism, chemical structure transformation maps, metabolism of cofactors and vitamins, xenobiotics biodegradation and metabolism and biosynthesis of other secondary metabolites (Figure 7B). Based on a significance threshold of *p* ≤ 0.05, 20 metabolic pathways were selected. In the negative ion mode, the enriched pathways included biosynthesis of alkaloids derived from the shikimate pathway, biosynthesis of amino acids, 2-oxocarboxylic acid metabolism, secondary bile acid biosynthesis and pantothenate and CoA biosynthesis (Figure 7C). In the positive ion mode, the enriched pathways were steroid hormone biosynthesis, methane metabolism, insulin resistance, galactose metabolism, carbon metabolism and carbon fixation in photosynthetic organisms (Figure 7D).

### 3.6. Correlation Analysis of Microbiota, Enzyme Activities and Metabolites in Rumen Fluid

Among the microbial groups, the top 25 ranked genera with significant differences were selected as the differential core genera. The key metabolites involved in carbohydrate metabolism and energy metabolism were identified from the KEGG pathway analysis. Differential metabolites were then screened using a criterion of VIP > 1. Subsequently, correlation analyses were conducted among the microorganisms, enzyme activities, and screened differential metabolites. The results of the correlation analysis of microbiota and enzyme activities showed that *Clostridium_sensu_stricto_1*, *Corynebacterium*, *Desemzia*, *Lachnospiraceae_ND3007_group*, *Paeniclostridium*, *Psychrobacter*, *Quinella*, *Romboutsia*, *Turicibacter*, *[Eubacterium]_nodatum_group* and *hoa5-07d05_gut_group* showed positive correlations with amylase, butyrate kinase, cellulase, hemicellulase, pepsin, phosphofructokinase, pyruvate kinase, succinate dehydrogenase, sucrase and trypsin; while exhibiting negative correlations with acetate kinase and lactate dehydrogenase. In contrast, *Lachnospiraceae_AC2044_group*, *Lachnospiraceae_NK4A136_group*, *Lachnospiraceae_UCG-009*, *Lachnospiraceae_XPB1014_group*, *Ruminococcus*, *UCG-002*, *UCG-004* and *unidentified_F082* demonstrated positive correlations with acetate kinase and lactate dehydrogenase but negative correlations with the other enzymes (Figure 8A). The results of the correlation analysis of the enzyme activities and metabolites show that amylase, butyrate kinase, cellulase, pepsin, phosphofructokinase, pyruvate kinase, succinate dehydrogenase, sucrase and trypsin showed positive correlations with 2-isopropylmalate, citric acid, D-raffinose, fumaric acid, L-aspartic acid, tyramine and cis-aconitic acid; while exhibiting negative correlations with 2-hydroxybutyric acid, adenosine 5′-diphosphate, adenosine diphosphate (ADP), D-glucosamine 6-phosphate and phosphopyruvic acid. In contrast, acetate kinase and lactate dehydrogenase showed positive correlations with 2-hydroxybutyric acid, adenosine 5′-diphosphate, adenosine diphosphate (ADP), D-glucosamine 6-phosphate and phosphopyruvic acid, and negative correlations with the other metabolites (Figure 8B). The results of the correlation analysis of the microbiota and metabolites show that *Clostridium_sensu_stricto_1*, *Corynebacterium*, *Desemzia*, *Lachnospiraceae_ND3007_group*, *Paeniclostridium*, *Psychrobacter*, *Quinella*, *Romboutsia*, *Turicibacter*, *Tuzzerella*, *[Eubacterium]_nodatum_group* and the *hoa5-07d05_gut_group* showed positive correlations with 2-isopropylmalate, citric acid, D-raffinose, fumaric acid, L-aspartic acid, tyramine and cis-aconitic acid. On the other hand, *Lachnospiraceae_AC2044_group*, *Lachnospiraceae_UCG-009*, *Lachnospiraceae_XPB1014_group*, *Ruminococcus*, *UCG-002*, *UCG-004* and *unidentified_F082* were positively correlated with 2-hydroxybutyric acid, adenosine 5′-diphosphate, adenosine diphosphate (ADP) and phosphopyruvic acid (Figure 8C).

## 4. Discussion

### 4.1. Effects of Different Feeding Methods on Growth Performance of Yak Calves

The growth and development of animals depend on the digestion and absorption of nutrients from their diet. For ruminants, the ratio of concentrate to roughage is particularly critical, as it determines the nutritional quality of the diet, which influences growth performance [22]. High-concentrate diets can supply dense nutrients, meeting the maintenance and production requirements of ruminants and thereby enhancing productivity [23,24]. Body size indices are positively correlated with body weight and can reflect the developmental status of animals [25]. The present study revealed that calves in the CO group performed better in terms of wither height, chest circumference, ADG and total body weight gain and improved in terms of structural measurements. A previous study with an F:C ratio of 3:7 reported that 8-month-old male yaks showed improved growth performance, including ADG and total body weight gain on the Qinghai-Tibetan Plateau [26]. Prolonged milk feeding in dairy calves predisposes them to lesser solid dry matter intake and, consequently, poor rumen development and delayed weaning as well as delayed onset of puberty [10,27]. High-concentrate diets also improve daily feed intake and slaughter performance, enhance muscles’ water-holding capacity and meat color, increase the variety of hydrocarbon compounds and the content of high-density lipoprotein cholesterol and reduce muscle lipid oxidation. However, the effect of this dietary transition on the rumen tissue development and muscle flavor of young yaks remains inconspicuous [16]. As calves are born with immature rumen, solid intake is the most important and demanding requirement for the establishment of a rumen microbial community and to develop a fully functional rumen [28]. For a smooth transition from liquid to solid, a proper feeding strategy should be adopted that stimulates and encourages solid feed intake; otherwise, calves will stick to milk suckling [29]. Another disadvantage of prolonged suckling is delaying the resumption of estrus cycling in female yaks. Suckling is the major factor influencing the resumption of postpartum ovarian cycles, which affect hypothalamic, pituitary and ovarian activity and thus inhibit follicular development [30]. Therefore, early weaning could theoretically improve the reproductive rate of yaks, enhancing the marketability of yak calves by bringing them to market faster and at a more desirable weight. However, the sample size, feeding behavior, health benefits for the consumer and long-term ecological impacts such as greenhouse gas emissions require further and more thorough evaluation.

### 4.2. Effects of Different Feeding Methods on VFA Content of Yak Calves

The rumen serves as the primary natural fermentation system in ruminants, exhibiting the highest capacity for degrading fibrous materials. It hosts a diverse microbial community comprising bacteria, fungi and protozoa, among others [31]. These microorganisms digest plant-derived feed components, such as cellulose and hemicellulose, which are otherwise indigestible by the host and convert them into short-chain fatty acids (SCFAs) like acetate and propionate. These SCFAs are subsequently utilized by the host as key sources of energy and essential nutrients [32]. The present study observed significantly higher concentrations of acetate, propionate, butyrate and total VFAs in the rumen fluid of the CO group compared with the SU group. VFAs produced by carbohydrate degradation are the primary energy source for ruminants. Since the concentrate feed contains a higher proportion of fermentable carbohydrates, the CO group exhibited a higher VFA content. In a comparative study on yaks and Qaidam cattle fed diets with varying energy levels, Liu et al. [33] observed a linear increase in ruminal VFAs with increasing dietary metabolizable energy. Hao et al. [34] investigated rumen fermentation in dairy cows during the pre-weaning and post-weaning periods and observed that the concentrations of acetate, propionate, butyrate and total volatile fatty acids were significantly higher under post-weaning conditions. Although the acetate-to-propionate ratio did not decrease significantly, it was slightly lower in the CO group, which may be attributed to factors such as calf physiology, the duration of feeding and dietary composition.

### 4.3. Effects of Different Feeding Methods on Activitives of Digestive Enzymes

Digestive enzymes are vital biological substances in animals, serving as the foundation for the digestion and absorption of dietary nutrients. The activity of these enzymes can serve as an indirect indicator of an animal’s capacity for nutrient digestion and absorption. The rumen hosts a diverse microbial community, primarily composed of bacteria, archaea, fungi and protists [35]. These microorganisms secrete a variety of digestive enzymes, such as cellulase, hemicellulase and pectinase, that break down complex plant polymers including proteins, lipids and carbohydrates, thereby facilitating the host’s nutrient absorption and energy acquisition [36,37]. Diet plays a key role in shaping the composition and abundance of rumen microorganisms, which in turn influences the activity of microbial digestive enzymes and indirectly affects overall digestive enzyme activity [38]. Indigestible dietary components like cellulose, hemicellulose and lignin are degraded by cellulase into glucose and VFAs. Cellulase activity is closely associated with the concentrate-to-roughage ratio in the diet and is commonly used as an indicator of the fiber-degrading capacity of rumen microbes [39,40]. In this study, we observed that cellulase activity was significantly higher in the CO group compared with the SU group. This suggests that dietary supplementation with concentrated feed and oat green hay directly enhances rumen digestive enzyme activity. Oat green hay, being rich in cellulose, appears to stimulate fiber-degrading microorganisms to produce enzymes that facilitate the breakdown of crude fiber. Some studies have indicated that the activities of protease, carboxymethyl cellulase, cellobiose hydrolase and β-glucosidase tend to be lower in weaned calves compared with pre-weaned calves, a finding that contrasts with the results of the present study [34]. This discrepancy may be attributed to differences in feeding duration. There is also a synergistic relationship between cellulase activity and the VFA concentration; higher cellulase activity reflects a stronger microbial ability to degrade cellulose, leading to increased VFA production [41], which aligns with the measured VFA levels. Moreover, the proportion of concentrate in the diet influences amylase activity. High-concentrate diets provide ample amounts of non-fiber carbohydrates, promoting the proliferation of starch-degrading bacteria and thereby enhancing the activity of amylase and protease [42]. These findings are consistent with the results in the present study, indicating that high-concentrate feeding strategies can boost the activity of fiber-degrading enzymes and promote greater energy production.

### 4.4. Effects of Different Feeding Methods on Rumen Microbial Community Composition

The structure and diversity of rumen microbial communities are closely associated with factors such as the host’s species, age and dietary composition. Among these, diet exerts the most significant influence on the rumen microbial flora. Variations in the concentrate-to-roughage ratio lead to corresponding shifts in the structure, composition and abundance of microbial communities [43,44]. An increase in the proportion of concentrate feed in the daily ration has been shown to reduce rumen microbial diversity [45]. The present study revealed notable differences in ASV composition between the two experimental groups. The CO yak calves yielded lower ASVs numbers, Shannon index values and Chao1 index values for the rumen microbiota compared with the SU group, indicating reduced microbial richness and diversity. Benchaar et al. [46] reported that as the concentrate level in the feed increased, both the diversity and richness of the rumen microbiota in dairy cows decreased significantly. Similarly, Hu et al. [47] observed that high-energy feed reduced the abundance and diversity of rumen bacteria in yaks. Furthermore, PCoA analysis demonstrated that the dietary concentrate-to-roughage ratio also influenced the structure of the rumen microbiota, which is consistent with the findings reported by Islam [48].

Bacteroidetes and Firmicutes are the dominant bacterial phyla in the rumen of yaks, collectively representing over 90% of the total rumen bacterial community [17]. Bacteroidetes primarily function in carbohydrate degradation, breaking down dietary energy and protein sources into absorbable small molecules. Firmicutes, on the other hand, produce a variety of digestive enzymes that degrade fibrous materials and convert them into VFAs and other metabolites, thereby facilitating the host’s nutrient digestion and absorption. Both phyla play crucial roles in the nutritional metabolism of ruminants [49]. Yaks fed high-protein diets exhibit a relatively high abundance of Bacteroidetes, which enhances the digestion and absorption of dietary proteins and carbohydrates [50]. In the present study, Bacteroidetes and Firmicutes were identified as the dominant phyla in the yak rumen. In the CO group, the abundance of Bacteroidota was higher than that of Firmicutes. The CO group’s diet resulted in an increased relative abundance of Bacteroidetes. In contrast, the opposite pattern was observed in the SU group. Ahmad et al. [51] reported that high-energy feeding increases the relative abundance of Firmicutes and Bacteroidetes in the cattle rumen, while an increased proportion of concentrate versus forage in the diet leads to a gradual decrease in Firmicutes. Conversely, Chen et al. [52] found that a higher hay content in the feed promotes the abundance of Firmicutes. The higher relative abundance of Firmicutes observed in the SU group may be attributed to the calves’ consumption of a cow’s diet during the later stage of the experiment.

At the genus level, the rumen microbial community is predominantly represented by *Prevotella* from the phylum Bacteroidetes, which plays a crucial role in degrading ruminal polysaccharides. By associating with fiber-degrading bacteria, *Prevotella* breaks down starch, protein, xylan and pectin from plant materials and participates in dietary fiber fermentation and VFA production, thereby regulating protein and carbohydrate metabolism [53]. This study found that CO feeding increased the relative abundance of *Rikenellaceae_RC9_gut_group*, *Christensenellaceae_R-7* and *Prevotella* but decreased that of *Succiniclasticum* and *Ruminococcus*. Higher *Prevotella* levels in the CO group could be associated with solid feed intake. As previously reported, dietary composition influences the relative abundance of *Prevotella* in the rumen; its abundance increases with a higher proportion of concentrate in the diet [54]. Both *NK4A214* and *Christensenellaceae_R-7* belong to the phylum Firmicutes and contribute significantly to the degradation of cellulose and hemicellulose in the rumen [55]. High-concentrate diets reduce the abundance of *Christensenellaceae_R-7* and *NK4A214*, while increasing the relative abundance of *Succiniclasticum* and *Rikenellaceae_RC9_gut_group*, thereby enhancing starch and sugar degradation. Additionally, the abundance of *Rikenellaceae_RC9_gut_group* is positively correlated with VFA and acetic acid concentrations [56]. *Ruminococcus* is strongly associated with fiber degradation; the short-chain fatty acids produced by its metabolic activity can be utilized by the host, and their relative abundance rises with an increasing concentrate-to-forage ratio [57]. Diets rich in concentrate reduce diversity and increase fermentation efficiency. Some different results may be linked to the transition of calves to an adult-type diet.

LEfSe analysis revealed that the Bacteroidetes phylum was enriched in the CO group, whereas the SU group showed a higher abundance of Firmicutes. At the class level, the CO group was enriched with Bacteroidia, while the SU group was dominated by Clostridia. At the order level, the CO group was enriched with Bacteroidales, while the SU group exhibited increased abundances of Oscillatoriales and Trichospirales. At the family level, the CO group was enriched with Muribaculaceae, while the SU group showed higher levels of Ruminococcaceae and Helicobacteraceae. The Bacteroidetes phylum primarily degrades non-fibrous carbohydrates and is involved in the breakdown of non-fibrous carbohydrates and proteins, whereas the Firmicutes phylum specializes in the decomposition of fibrous materials [58]. Helicobacteraceae and Ruminococcaceae are capable of degrading cellulose and hemicellulose in feed to produce volatile fatty acids, a process influenced by the concentrate-to-forage ratio in the diet [58,59]. The unexpected increase in the abundance of a microbial community could be associated with the transfer of microbes from the mother through the consumption of yaks’ saliva-contaminated feed.

### 4.5. Effects of Different Feeding Methods on Rumen Fluid’s Untargeted Metabolome

Metabolomics can elucidate the metabolic interactions between the rumen microbial community and the host. Changes in metabolite profiles reflect the involvement of dietary nutrients in microbial metabolism within the rumen. Studies have shown that the concentration and metabolic pathways of rumen metabolites in yaks are influenced by the concentrate-to-forage ratio in the diet [60]. Principal component analysis revealed a significant difference in metabolite composition between the CO and SU groups, suggesting that high-concentrate feeding alters the rumen metabolome. Furthermore, metabolomic analysis indicated that different feeding regimens significantly affect metabolic pathways such as 2-oxocarboxylic acid metabolism, steroid hormone biosynthesis and amino acid biosynthesis in the rumen fluid of young yaks.

The 2-oxocarboxylic acid metabolic pathway comprises various biochemical reactions involving 2-oxocarboxylic acids and represents a significant category of metabolic pathways in living organisms. Metabolites within this pathway play crucial roles in energy metabolism, amino acid synthesis and degradation and the biosynthesis of secondary metabolites. They are also implicated in immune regulation processes [61,62]. For instance, 2-oxovaleric acid can be converted into glutamate, linking it to the arginine biosynthesis pathway to produce N-acetylornithine, and 2-oxoadipic acid participates in lysine biosynthesis, while 2-oxoisovaleric acid is transformed into 2-isopropylmalic acid in the presence of acetyl-CoA and contributes to the biosynthetic pathways of valine, leucine and isoleucine. Steroid hormones constitute a group of diverse signaling molecules, including glucocorticoids, mineralocorticoids, androgens, estrogens and progesterone. They function as signaling molecules within the body, regulating the water-electrolyte balance and glucose-lipid metabolism [63]. Research has shown that metabolites involved in steroid hormone biosynthesis are widely present in the ruminant digestive tract and can serve as biomarkers to distinguish microbial community features across different gastrointestinal regions [64]. Owing to their lipophilic nature, steroid hormones readily cross cell membranes to exert their functions. They bind to intracellular receptors and directly modulate gene expression, thereby influencing metabolic processes, including the metabolism of carbohydrates and other substances [65]. However, the risks of high-concentrate diets associated with nutrient metabolism disorders in ruminants require further evaluation.

### 4.6. Effects of Different Feeding Methods on Correlation of Microbiota, Enzyme Activities and Metabolites

The rumen harbors a diverse array of microorganisms. These microbial communities secrete various digestive enzymes that break down nutrients in feed into smaller molecules, which are then absorbed by the host [36,41,66]. Through modulating enzyme activity, these microbes regulate metabolite production. For instance, *Clostridium* species effectively degrade cellulose by regulating the activity of cellulases and hemicellulases [67], and lactate dehydrogenases secreted by certain microorganisms help promote propionate synthesis [68]. Our analysis revealed positive correlations among specific microbial genera (*Clostridium_sensu_stricto_1*, *Corynebacterium*, *Desemzia*, *Lachnospiraceae_ND3007_group*, *Paeniclostridium*, *Psychrobacter*, *Quinella*, *Romboutsia*, *Turicibacter*, *[Eubacterium]_nodatum_group* and *hoa5-07d05_gut_group*), enzyme activities (amylase, butyrate kinase, cellulase, hemicellulase, pepsin, phosphofructokinase, pyruvate kinase, succinate dehydrogenase, sucrase and trypsin) and metabolites (2-isopropylmalate, citric acid, D-raffinose, fumaric acid, L-aspartic acid, tyramine and cis-aconitic acid). In addition, positive correlations were also observed between another set of microbial genera (*Lachnospiraceae_AC2044_group*, *Lachnospiraceae_NK4A136_group*, *Lachnospiraceae_XPB1014_group*, *Ruminococcus*, *Lachnospiraceae_UCG-009*, *UCG-002*, *UCG-004* and *unidentified_F082*), enzyme activities (acetate kinase and lactate dehydrogenase) and metabolites (2-hydroxybutyric acid, adenosine 5′-diphosphate, D-glucosamine 6-phosphate and phosphopyruvic acid). Due to its uniform composition, a high-concentrate diet reduces the diversity of rumen microorganisms. This alteration in microbial diversity further leads to changes in metabolites. However, the high nutritional content of the high-concentrate diet contributes to energy production, and the activity of enzymes involved in energy digestion and metabolism also increases accordingly. Given that many of the differing microbial genera were not core taxa in this study, our analysis was limited to a preliminary correlation assessment. The potential relationships between these microorganisms and enzyme activities, as well as metabolites, warrant further experimental validation.

## 5. Conclusions

In summary, compared with suckling a dam, early weaning with full feeding improved the heights and chest circumferences of the calves, along with a 33.28% increase in their final body weights. It also significantly enhanced rumen fermentation, indicated by the elevated levels of VFAs and increased fibrolytic enzyme activity in the rumen fluid. Additionally, this feeding strategy reduced the diversity and richness of rumen microorganisms, characterized by an increased relative abundance of Bacteroidetes and decreased abundance of Firmicutes and *Ruminococcus*, while also altering metabolite profiles. These findings provide practical insights into the healthy rearing of yak calves born in June and July. Future research should focus on elucidating the mechanisms of energy absorption and utilization, as well as evaluating the effects of high-concentrate feeding on rumen acidosis, economic benefits and ecological sustainability.

## Figures and Tables

**Figure 1 microorganisms-14-00081-f001:**
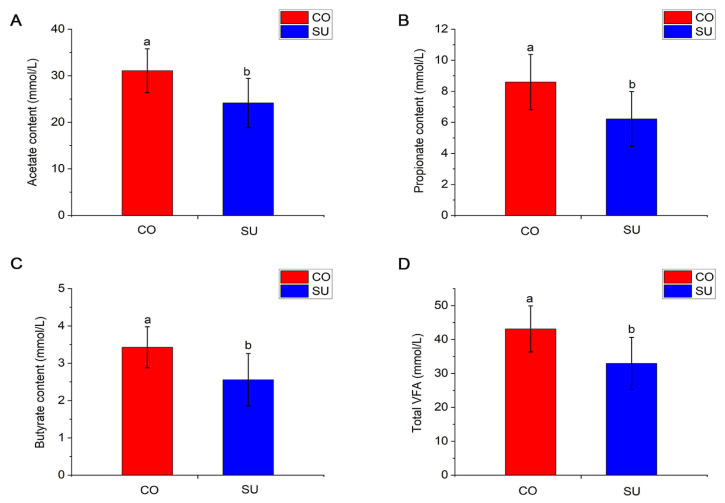
The influence of different feeding methods on the rumen fluid VFA content of yak calves. (**A**) Acetate content. (**B**) Propionate content. (**C**) Butyrate content. (**D**) Total VFA content. Different lowercase letters indicate significant differences (*p* < 0.05).

**Figure 2 microorganisms-14-00081-f002:**
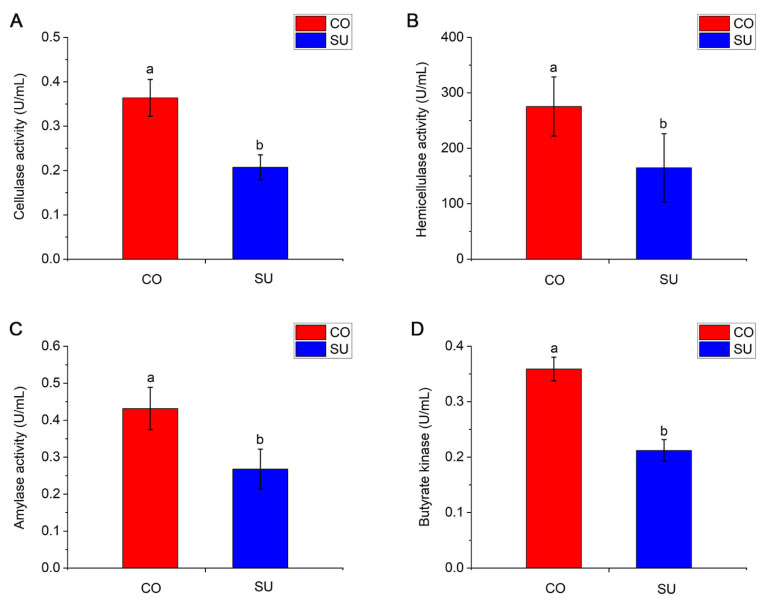
The influence of different feeding methods on the rumen fluid enzyme activities of yak calves. (**A**) Cellulase activity. (**B**) Hemicellulase activity. (**C**) Amylase activity. (**D**) Butyrate kinase activity. Different lowercase letters indicate significant differences (*p* < 0.05).

**Figure 3 microorganisms-14-00081-f003:**
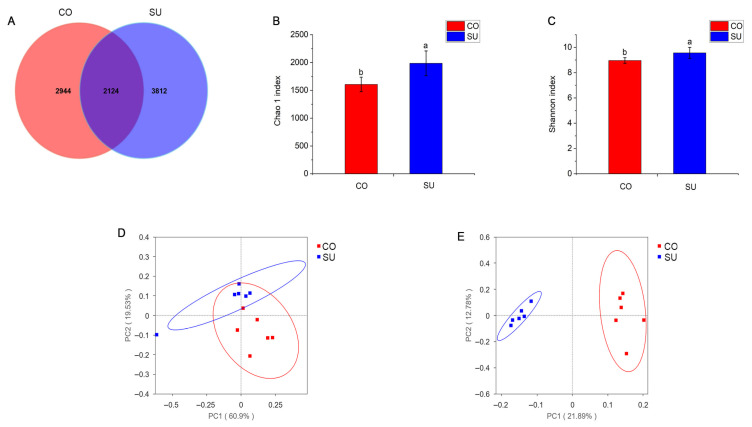
The influence of different feeding methods on the rumen microbial diversity of yak calves. (**A**) Venn diagram. (**B**) Chao 1 index. (**C**) Shannon index. (**D**) Based on the weighted unifrac distance PCoA graph. (**E**) Based on the unweighted unifrac distance PCoA graph. Different lowercase letters indicate significant differences (*p* < 0.05).

**Figure 4 microorganisms-14-00081-f004:**
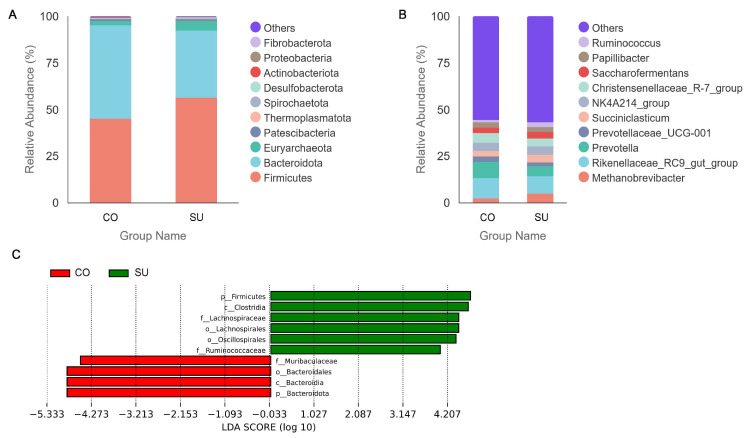
Relative abundance of rumen microbiota in yak calves suckling or fed a high-concentrate diet. (**A**) Relative abundance of rumen microbiota phyla. (**B**) Relative abundance of rumen microbiota genera. (**C**) Histogram of LDA topic distribution.

**Figure 5 microorganisms-14-00081-f005:**
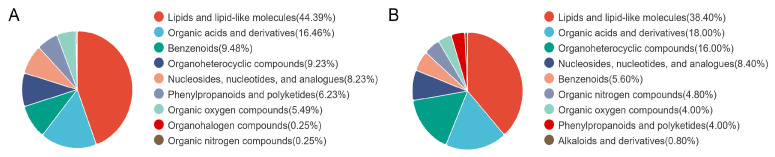
Qualitative and quantitative classification of rumen metabolites. (**A**) Negative ion regulation. (**B**) Positive ion regulation.

**Figure 6 microorganisms-14-00081-f006:**
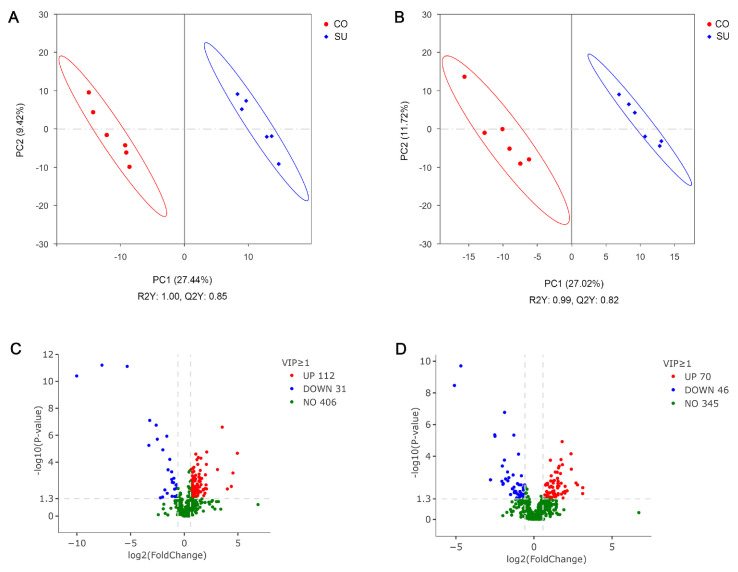
Differential metabolite screenings in rumen fluid of suckling and high concentrate-fed yak calves. (**A**) PLS-DA score plot in negative ion regulation. (**B**) PLS-DA score plot in positive ion regulation. (**C**) Volcano plots in negative ion regulation. (**D**) Volcano plots in positive ion regulation.

**Figure 7 microorganisms-14-00081-f007:**
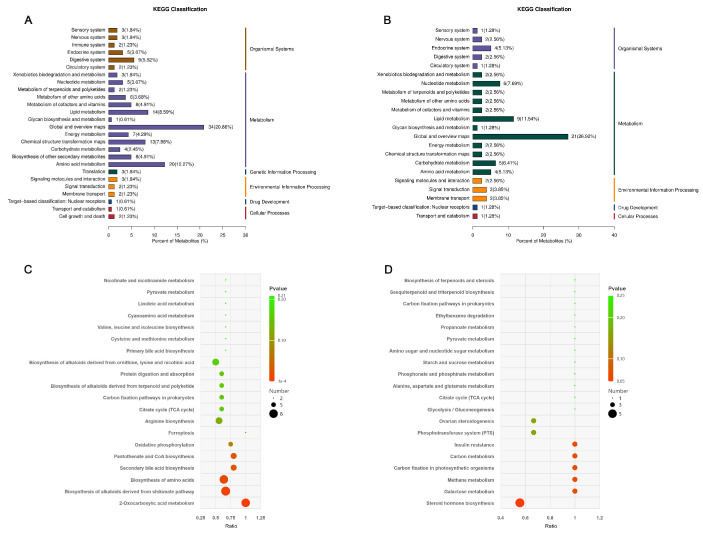
Functional annotation and enrichment analysis of KEGG for differential metabolites. (**A**,**C**) Negative ion regulation. (**B**,**D**) Positive ion regulation.

**Figure 8 microorganisms-14-00081-f008:**
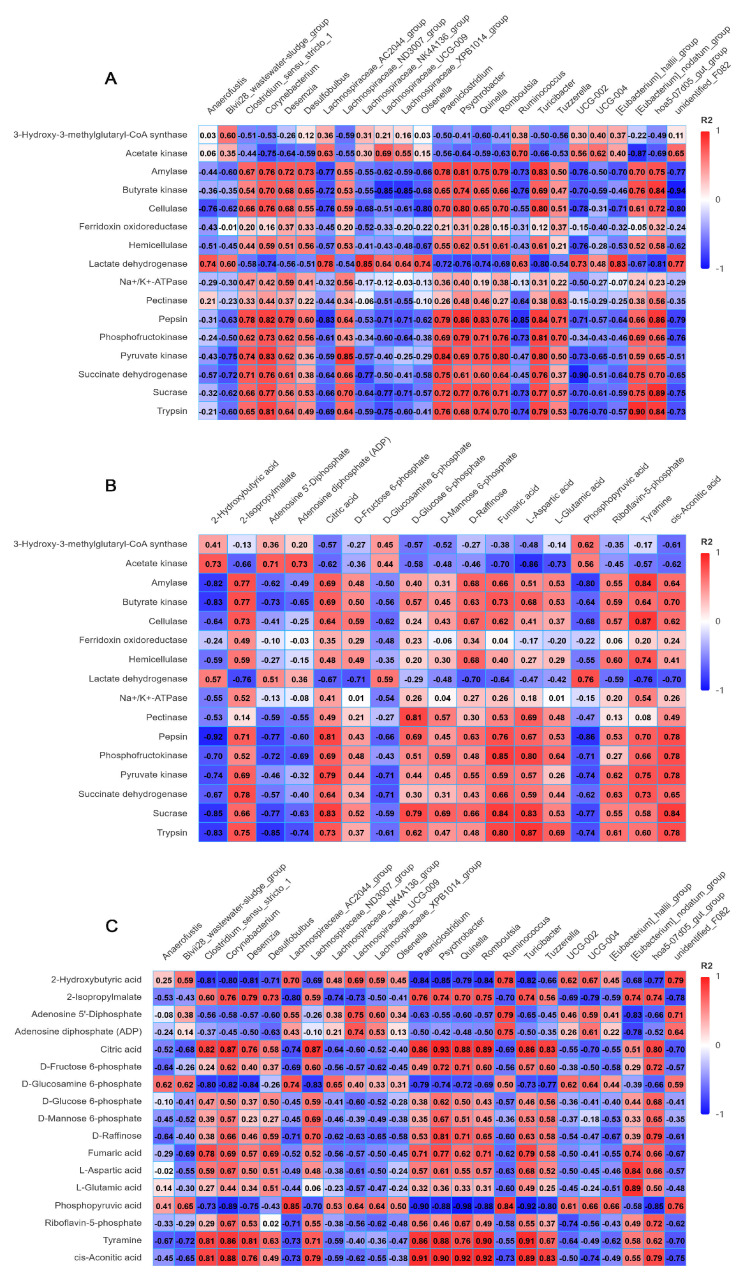
Correlation analysis of microbiota, enzyme activities and metabolites. (**A**) Correlation analysis of microbiota and enzyme activities. (**B**) Correlation analysis of enzyme activities and metabolites. (**C**) Correlation analysis of microbiota and metabolites.

**Table 1 microorganisms-14-00081-t001:** Composition and nutrient levels of the experimental diet (dry matter basis (%)).

Feed Composition	Content
Oat hay	30.00
Corn	37.10
Wheat bran	8.40
Soybean meal	10.50
Canola meal	8.40
CaHPO_4_	1.05
CaCO_3_	1.05
NaHCO_3_	0.70
NaCl	2.10
Premix ^1^	0.70
Total	100.00
Nutrient levels ^2^	
Metabolic energy ^3^, MJ/kg	9.02
Crude protein	12.78
Ether extract	1.68
Ash	7.79
Neutral detergent fiber	29.03
Acid detergent fiber	18.87
Ca	0.51
P	0.54

^1^ The premix provided the following per kg for the diet: vitamin A 5000 IU, vitamin D_3_ 1000 IU, vitamin E 50 IU, Cu 10 mg, Fe 60 mg, Mn 25.5 mg, Zn 25 mg, I 0.75 mg, Se 0.45 mg, Co 0.29 mg. ^2^ Nutrient levels were measured values. ^3^ ME is a calculated value [19], where ME (MJ/kg) = 9.236 − 0.213 Ash + 0.044 CP + 0.300 EE + 0.020 ADF.

**Table 2 microorganisms-14-00081-t002:** Body weight and body measurements (mean ± SEM) of yak calves at 1st day and 120th day of the experiment.

Parameters	CO	SU	*p* Value
Initial wither height, cm	78.33 ± 4.68	82.33 ± 4.59	0.166
Final wither height, cm	92.17 ± 2.23 ^a^	85.50 ± 1.87 ^b^	<0.001
Initial body length, cm	76.50 ± 4.76	82.00 ± 6.78	0.135
Final body length, cm	89.83 ± 6.31	86.17 ± 7.19	0.370
Initial chest circumference, cm	111.33 ± 6.12	111.83 ± 4.96	0.880
Final chest circumference, cm	130.33 ± 3.88 ^a^	115.33 ± 7.87 ^b^	0.002
Initial weight, kg	67.43 ± 7.44	69.63 ± 5.68	0.577
Final weight, kg	105.93 ± 8.74 ^a^	79.48 ± 6.89 ^b^	<0.001
Total weight gain, kg	38.49 ± 4.51 ^a^	9.85 ± 4.03 ^b^	<0.001
ADG, kg/d	0.32 ± 0.04 ^a^	0.08 ± 0.03 ^b^	<0.001

Different lowercase letters for means in a row indicate significant differences (*p* < 0.05).

## Data Availability

The original contributions presented in this study are included in the article/Appendix A. Further inquiries can be directed to the corresponding authors.

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
