# Peer review of "Effects of Different Feeding Methods on Growth Performance, Enzyme Activity, Rumen Microbial Diversity and Metabolomic Profiles in Yak Calves"

_microorganisms, 2025, doi:10.3390/microorganisms14010081_

Round 1

Reviewer 1 Report

Comments and Suggestions for Authors

The work by Wang et al focuses on understanding how different weaning and feeding strategies can impact the growth performance of yak calves and, consequently, the production cycle of this species. By using metabolomics and 16S rRNA Gene Sequencing, the authors show that providing a concentrate-based diet enhances the growth performance of yak calves, elicited by the increase in ruminal volatile fatty acids and cellulase activity—both of which boost the metabolizable energy. Consequently, the diversity and composition of the ruman microbial population was altered by diet which was reflected by the metabolic profile of the ruminal fluid. 

Major Comments:

While this is a well-designed study, with solid conclusions, the novelty of this study is somewhat lacking. More importantly, the authors conclude the paper without discussion the limitations of their study or the downside of accelerating the growth of yak calves. For instance, will a concentrate-based diet impact the quality (flavor profile, health benefits for the consumer, etc) of the meat of yak calves? What are the economical implications for the yak producers? Is a concentrate-based diet profitable? What are the possible consequences for the environment? Bottom line: Is this a realistic and sustainable feeding strategy?

Minor Comments:

Line 132: specify which internal standard was used.
Line 151: explain the acronym TMB.
Line 175: specify the sample injection volume.
Tables 3, 4, and 5: move tables to supplementary materials and show bar plots with individual datapoints or violin plots in the main text to help reader visualize the results better.
Tables S3 and S4: Include column with metabolite name instead of using legend/footnote; as is, it is hard to read and comprehend.

Reviewer 2 Report

Comments and Suggestions for Authors

Suggestions and corrections in manuscript:

Adjust the title:“Effects of Prolonged Suckling and Early Weaning on Growth, Rumen Microbiota, Enzyme Activity, and Metabolomic Profiles in Yak Calves”

ABSTRACT

Line 22-24 “Traditional natural feeding only prolongs suckling period, which often lead to a series of problems, constrained by the harsh high altitude environment, such as inadequate nutrition, retarded growth, and imbalanced herd structure.” replace by “Exclusive traditional natural feeding prolongs the suckling period, and this leads to a series of problems due to the harsh high-altitude environment, such as inadequate nutrition leading to retarded growth and an imbalanced herd structure.”

I suggest rewriting is deprived of the summary in the most direct and short way.

Line 30 replace "complete weaning" by "early weaning with full feeding".

Inserir de forma clara e direta o objetivo da pesquisa.

INTRODUTION

The text needs to be improved, lacking fluidity and coherence; there are disconnected phrases, sudden changes of theme, reorganization in thematic blocks (Importance of yak; Problem of prolonged breastfeeding; Consequences for ruminal development; Knowledge gaps; Study objective).

Line 52-53 replace "... meet your needs that cannot be met..." by "... providing essential resources that other animals cannot provide in such harsh conditions."

Line 56 replace "...is critical period..." by "a critical period".

Line 60 replace "agricultural system" by "agricultural systems".

Insert a clear and direct sentence of the manuscript goal, for example:

“Therefore, this study aimed to compare prolonged suckling and early weaning with total‑pen feeding in yak calves, focusing on growth performance, rumen fermentation, microbial diversity, and metabolomic profiles.”

MATERIALS AND METHODS

Line 91: It is important to describe what the herd total is and how many calves were removed to remove the 12 “... were selected for the study…" thus characterizing how sampling was performed.

Line 92-97: confusing description of treatments and food management, I suggest rewriting directly. For example, in line 95, replace "they can eat the cow’s food freely" by "SU group calves suckled ad libitum and had access to the same forage consumed by their dams."

Lack of essential information: nutritional composition of breast milk (even if not measured, it should be recognized as a limitation); individual dry matter intake (IDM); environmental conditions during the experiment (temperature, humidity, seasonal variations).

Table 1 shows the composition of concentrate and hay, but does not show the final composition of the 70:30 diet. I suggest adding an additional column with the final composition. In the same table 1, on line 112, set "(air-dry basis, %)." for "(dry matter basis, %).

There is no description of the calculation of ADG (Average Daily Gain),  although it is simple, and should be explained. In the same sense, there is no mention of pre-weighing management (fasting, time without water). For example, "ADG was calculated as (final BW - initial BW) / 120 days.".

Line 122: Method of euthanasia not described, and there is no description of the time between euthanasia and the collection of ruminal fluid and freezing.

There is no mention of ruminal pH. pH is a fundamental parameter and should be included.

The description of VFA determination is too detailed for instrumental parameters, but essential information is missing: standard curve, detection limits, quality control, analytical replicates, and there is no mention of the use of a specific internal standard.

There is no complete list of enzymes evaluated in the methodology, but they appear in Table 4.

Line 156 - 170: no description of the Illumina platform (MiSeq? NovaSeq?); there is no mention of the number of reads per sample; there is no description of contamination control.

Line 171-184 Metabolite Determination: no description of the ionization mode (positive/negative); no mention of the number of metabolites identified; no description of the normalization method.

DISCUSSION

The text alternates themes without clear transitions, which impairs fluidity. I suggest that the discussion follow a logical order: Growth; Ruminal fermentation; Enzymes; Microbiota; Metabolomics; Integration of findings; Limitations; Practical implications.

The authors state that the CO group had higher VFA, but do not explain biologically why this occurs (higher intake of fermentable carbohydrates, higher passage rate, greater microbial activity).

Lack of integration between microbiota, enzymes, and metabolomics. The results show: Reduction of microbial diversity; Increase of digestive enzymes; Important metabolic alterations. But the discussion does not integrate these three levels, which is a missed opportunity.

I believe that there was a misinterpretation of microbial diversity reduction, because the text treats diversity reduction as something negative, but in young animals, it is expected that diets rich in concentrate reduce diversity and increase fermentation efficiency. The literature shows that less diversity does not imply worse performance.

Absence of study limitations. For example, there could be points such as: small sample size (n=6 per group); absence of dry matter consumption measurement (DMI); absence of ruminal pH; absence of ruminal histology data; absence of feeding behavior data.

Lack of practical implications in the discussion, as the study has strong relevance for yak calves management. But the discussion does not explore aspects such as: economic impact; welfare impact; and applicability in extensive systems.

The manuscript has potential, but the current discussion does not adequately explore the wealth of data.

CONCLUSIONS

I suggest that the conclusions be adjusted to be short and direct, highlighting the most relevant points, such as the 33% increase in final weight that should be emphasized. Include practical implications, limitations, and future perspectives.
